# Recent Advance Analysis of Recovery in Hospitalized People with COVID-19: A Systematic Review

**DOI:** 10.3390/ijerph192114609

**Published:** 2022-11-07

**Authors:** Joyce Noelly Vitor Santos, Vanessa Amaral Mendonça, Amanda Cristina Fernandes, Laísa Braga Maia, Nicholas Henschke, Mateus Bastos de Souza, Vanessa Kelly da Silva Lage, Murilo Xavier Oliveira, Angélica de Fátima Silva, Ana Cristina Rodrigues Lacerda, Alessandro Sartorio, Amandine Rapin, Vinícius Cunha de Oliveira, Redha Taiar

**Affiliations:** 1Programa de Pós-Graduação em Reabilitação e Desempenho Funcional (PPGReab), Universidade Federal dos Vales do Jequitinhonha e Mucuri, Diamantina 39100-000, MG, Brazil; 2Programa de Pós-Graduação em Ciências da Saúde (PPGCS), Universidade Federal dos Vales do Jequitinhonha e Mucuri, Diamantina 39100-000, MG, Brazil; 3Institute for Musculoskeletal Health, School of Public Health, The University of Sydney, Sydney 2006, Australia; 4Programa de Pós-Graduação Multicêntrico em Ciências Fisiológicas (PPGMCF), Universidade Federal dos Vales do Jequitinhonha e Mucuri, Diamantina 39100-000, MG, Brazil; 5Istituto Auxologico Italiano, Istituto di Ricovero e Cura a Carattere Scientifico (IRCCS), Experimental Laboratory for Auxo-endocrinological Research, 20145 Milan, Italy; 6Département de Médecine Physique et de Réadaptation, Hôpital Sébastopol, Centre Hospitalo-Universitaire de Reims (CHU), 51092 Reims, France; 7Faculté de Médecine, Université de Reims Champagne-Ardenne, Vieillissement, Fragilité (VieFra), 51092 Reims, France; 8MATIM, Moulin de la Housse, Université de Reims Champagne Ardenne, 51687 Reims, France

**Keywords:** COVID-19, coronavirus, SARS-CoV-2, prognosis, systematic review

## Abstract

Introduction: COVID-19 is a public health emergency all around the world. Severe illness occurred in about 14% of patients and 5% of patients developed critical illness, but the prognosis for these patients remains unclear. Objective: To describe the prognosis in hospitalized adults with COVID-19. Methods: The MEDLINE, EMBASE, AMED, and COCHRANE databases were searched for studies published up to 28 June 2021 without language restrictions. Descriptors were related to “COVID-19” and “prognosis”. Prospective inception cohort studies that assessed morbidity, mortality and recovery in hospitalized people over 18 years old with COVID-19 were included. Two independent reviewers selected eligible studies and extracted the available data. Acute respiratory distress syndrome (ARDS) and multiple organ failure (MOFS) were considered as outcomes for morbidity and discharge was considered for recovery. The Quality in Prognosis Studies (QUIPS) tool was used to assess risk of bias. Analyses were performed using Comprehensive Meta-Analysis (version 2.2.064). Results: We included 30 inception cohort studies investigating 13,717 people hospitalized with COVID-19 from different countries. The mean (SD) age was 60.90 (21.87) years, and there was high proportion of males (76.19%) and people with comorbidities (e.g., 49.44% with hypertension and 29.75% with diabetes). Findings suggested a high occurrence of morbidity, mainly related to ARDS. Morbidity rates varied across studies from 19% to 36% in hospital wards, and from 13% to 90% in Intensive Care Units—ICU. Mortality rates ranged from 4% to 38% in hospital wards and from 8% to 51% in ICU. Recovery rates ranged up to 94% and 65% in hospital wards and ICU, respectively. The included studies had high risk of bias in the confounding domain. Conclusions: The prognosis of people hospitalized with COVID-19 is an issue for the public health system worldwide, with high morbidity and mortality rates, mainly in ICU and for patients with comorbidities. Its prognosis emphasizes the need for appropriate prevention and management strategies.

## 1. Introduction

COVID-19 is a worldwide public health emergency, caused by Severe Acute Respiratory Syndrome Coronavirus 2 (SARS-CoV-2) [1]. About 456,908,767 confirmed cases of contamination by SARS-CoV-2 have been recorded, with 6,041,077 deaths worldwide up to 14 March 2022 [2].

Presentations of SARS-CoV-2 infection range from asymptomatic to mild or moderate respiratory and non-respiratory symptoms and severe COVID-19 pneumonia [3]. Severe illness occurred in about 14% of patients and 5% of patients developed critical illness requiring intensive care or mechanical ventilation assistance [4]. Studies have associated the severity and the fatality of COVID-19 with risk factors such as older age and serious pre-existing diseases [5,6].

Studies have reported morbidity, mortality and recovery outcomes in COVID-19 inpatients, but prognostic studies are limited by study design, the definition of inception cohort and the heterogeneity of samples such as age group, comorbidities, countries’ characteristics and severity of patients’ illnesses. Taking this context into account, there is a need for a systematic review of high methodological quality to investigate the prognosis of people hospitalized with COVID-19. The aim of this systematic review of prospective longitudinal inception cohort studies was to investigate the prognosis of COVID-19 in people hospitalized regarding the outcomes of morbidity, mortality and recovery. Estimates were provided by country and severity (hospital ward or Intensive Care Unit—ICU), and the presence of comorbidities was explored. The hypothesis of the present study was that the occurrence of morbidity and mortality related to COVID-19 in hospitalized patients is high and might be impacted by the presence of comorbidities, hospital setting (severity) and countries with different health care systems.

## 2. Methods

### 2.1. Search Strategy and Selection Criteria

This systematic review was reported according to the PRISMA checklist and Cochrane Recommendations [7,8]. The protocol was registered prospectively in PROSPERO (CRD42021229355) and is available at https://www.crd.york.ac.uk/prospero/display_record.php?ID=CRD42021229355 and in Open Science Framework (https://doi.org/10.17605/OSF.IO/JG5DS), accessed on 22 January 2021.

The MEDLINE, EMBASE, AMED, and COCHRANE databases were searched for studies. No specific terms related to our outcomes of interest were used to increase the sensitivity of our search and avoid exclusions of possibly relevant studies. The detailed search strategy is available in Appendix A. We hand-searched reference lists of previous reviews in the area for potential full texts not identified by our searches.

All prospective inception cohort studies that assessed morbidity, mortality and recovery [9] in hospitalized people over 18 years old, with COVID-19 confirmed by laboratory test (RT-PCR of the naso-/oro-pharynx or serological test) with or without comorbidities, starting within ≤14 days from the onset of symptoms (i.e., an inception cohort) [3] were included. Core outcomes of acute respiratory distress syndrome (ARDS) and multiorgan failure (MOFS) were considered for morbidity [9]. An outcome of discharge from hospital was considered for recovery. Qualitative studies, retrospective studies, case reports, series, conference reports and comments, editorials and expert opinions were excluded.

### 2.2. Study Selection

After the searches, references were exported to an Endnote^®^ file and duplicates were removed. Two independent reviewers (JNS and ACF) screened titles and abstracts and assessed potential full texts for eligibility criteria. A third reviewer (LBM) resolved any between-reviewer disagreements. Three attempts to contact authors in order to clarify information were made.

### 2.3. Data Extraction

Two independent reviewers (JNS and ACF) extracted data from the included studies and a third reviewer (LBM) resolved disagreements. Author names, date of publication, type of study, city, country, sample source, sample size, patient comorbidities, inception cohort, description of treatment therapies, and hospital setting were extracted when available. Proportions of comorbidities at baseline, of morbidity, mortality and recovery were extracted.

### 2.4. Risk of Bias Assessment

Two independent reviewers (JNS and ACF) assessed the methodological quality of the included studies using the Quality in Prognosis Studies (QUIPS) modified tool [10,11,12]. The QUIPS tool assesses six domains: (i) study participation; (ii) study attrition; (iii) prognostic factor measurement; (iv) outcome measurement; (v) study confounding; and (vi) statistical analysis and reporting. Each domain was rated as having a high, moderate or low risk of bias. Disagreements were resolved by a third reviewer (LBM). Trained reviewers used a standardized form downloaded from the Cochrane Methods Prognosis website [13].

### 2.5. Data Analysis

The descriptive analyses and data summarization were performed using Comprehensive Meta-Analysis (version 2.2.064). For analyses, studies were grouped by country and hospital setting (hospital ward or ICU). Estimates of proportions for dichotomous data, considering the number of events and sample size, were reported. Planned subgroup analyses were not possible as planned because of data presentation, and sources of heterogeneity were descriptively explored.

## 3. Results

Searches identified 46,067 records and 29,767 titles and abstracts were screened after removing duplicates. Then, 255 potential full texts were assessed for our eligibility criteria and 30 studies investigating 13,717 patients hospitalized with COVID-19 were included in qualitative and quantitative synthesis. The main reasons for the exclusion of potential full texts were: not meeting the condition of interest (*n* = 149); not a prospective cohort (*n* = 61); not a journal paper (*n* = 14); and duplicates (*n* = 1) (see flow of studies in Figure 1 and reasons for exclusion of potential full texts in Appendix A).

### 3.1. Study Characteristics

Characteristics of patients and included studies are shown in Appendix A. Patients were hospitalized in a hospital ward [14,15,16,17,18,19,20,21,22,23,24,25,26,27] or ICU [28,29,30,31,32,33,34,35,36,37,38,39,40,41,42,43] and studies were conducted in different countries (Andorra [38], Belgium [29], Brazil [14], China [15,16], Czech Republic [28], Denmark [17], France [18,19,20,29], India [30], Italy [31], Mexico [22,23,32], Norway [24], Poland [25], Spain [26,33,34,35,36,37,38], Sweden [39], Switzerland [29], UK [40], USA [27,41,42,43]). All included studies were conducted in the first half of 2020. Of the 13,717 investigated patients, 4325 (31.53%) patients were hospitalized in a hospital ward on study admission, with 1407 (32.53%) of them transferred to ICU; and 9392 (68.46%) were hospitalized in ICU on study admission (see Appendix A). The mean (SD) age of the sample was 60.90 (21.87) years and 10,452 (76.19%) were male. The sample had a diagnosis of hypertension (49.44%), diabetes (29.75%), smoking history (20.37%), obesity (13.33%), chronic kidney disease (CKD) (12.15%), cancer (7.26%), asthma (7.20%), chronic obstructive pulmonary disease (COPD) (6.83%) and immunodeficiency (5.98%) at baseline. The prevalence of comorbidities is explored in Appendix A. Patients received heterogeneous categories of antibiotics and antivirals, in addition to clinical treatments such as oxygen therapy, high-flow nasal cannula, extracorporeal membrane oxygenation (ECMO), non-invasive ventilation and mechanical ventilation whilst hospitalized.

### 3.2. Risk of Bias

The risk of bias of the included studies is reported in Appendix A. We found a high risk of bias in terms of potential confounders not appropriately being addressed and adjusted for. The risk of bias in terms of study attrition was moderate, with most studies not presenting data on loss to follow-up. Although the risk of bias in terms of measurement of outcomes was low, 20% of the included studies were considered to have a moderate risk of bias in outcome measurement due to not reporting follow-up adequately. The risk of bias in terms of patient participation was moderate with most studies not presenting data on sample calculation and exclusion criteria. Regarding the domain of statistical analysis and report, the risk of bias was considered moderate. The risk of bias in individual studies is reported in Appendix A.

### 3.3. Summary of Evidence

A descriptive analysis was performed for each outcome of interest. Pooling was not estimated due to the heterogeneity across the studies. Prospective inception cohorts that assessed outcomes of mortality, morbidity (ARDS and MOFS) and recovery are reported in Figure 2, Figure 3, Figure 4 and Figure 5, respectively, considering hospital setting and countries where the studies were conducted.

In people hospitalized in wards, mortality estimates ranged from 7% to 38% across studies and were higher in Mexico (38%) and Brazil (30%), with lower estimates in Denmark (7%) and China (4%). In ICU, there were higher mortality estimates overall, ranging from 8% to 51%. Most studies reported a mortality rate of over 30% in ICU, with higher rates in Mexico (51%), Spain (51%), USA (42%) and Czech Republic (40%), and lower rates in India (8%) and UK (14%).

Occurrences of morbidity related to ARDS ranged from 19% to 36% across studies in hospital wards and from 13% to 98% in ICU. People hospitalized in ICU from France (98%), Spain (93%) and Czech Republic (77%) had higher occurrences of ARDS, whereas one study conducted in the USA reported lower rates of ARDS (13%). Regarding morbidity related to MOFS, occurrences ranged from 8% to 32% across studies in hospital wards and from 6% to 22% in ICU.

People hospitalized in hospital wards showed better prognosis; i.e., higher proportion of recovery (discharge to home), with estimates ranging from 55% to 94%. In ICU, recovery rates ranged from 22% to 76%; with higher recovery rates in Italy (55%), Spain (65%) and UK (76%). See Figure 5 for further details.

We also conducted descriptive analyses to explore whether the presence of comorbidities at baseline might impact the prognosis. High prevalence of comorbidities was found in people hospitalized with COVID-19 at baseline, mainly hypertension (rate up to 82% in hospital wards and ICU) (Appendix A), diabetes (up to 37% and 65% in hospital wards and ICU, respectively) (Appendix A), obesity (up to 44% and 82% in hospital wards and ICU, respectively) (Appendix A) and smoking history (up to 47% and 30% in hospital wards and ICU, respectively) (Appendix A). Hypertension had higher prevalence in studies conducted in USA (82%), Brazil (55%), Czech Republic (64%), Spain (59%) and France (53%); and diabetes had higher prevalence in USA (65%), Spain (41%), Czech Republic (40%) and Brazil (35%).

## 4. Discussion

The present systematic review showed that people hospitalized with COVID-19 have negative morbidity and mortality outcomes, despite a proportion of the sample recovered. Notably, this was the first systematic review based on a rigorous methodological design with the inclusion of prospective cohort studies and with a defined inception cohort, following Cochrane Recommendations [8,13], carried out in order to describe the prognosis in morbidity, mortality, and recovery outcomes in people hospitalized with COVID-19.

The inclusion of studies that presented the inception cohort defined was carried out with the aim of reducing bias related to the heterogeneity of the sample, considering the clinical course, including participants who are at an initial point of the disease and as uniform as possible. In addition, all studies were conducted in the first half of 2020, which reduces sample heterogeneity and discards heterogeneity related to variants and vaccination. Differences for occurrences across studies may be associated with different management strategies in different countries, the expertise of intensive healthcare workers, the heterogeneity of comorbidities, in addition to methodological factors such as sample size.

COVID-19 is a recent disease and for this reason, the scientific literature is still advancing in knowledge about its clinical course and prognosis. Many studies have been published reporting the disease course, but most do not have a methodological design that can describe the prognosis in relation to time (e.g., short, medium and long term) and not have high methodological rigor. In this sense, a descriptive analysis of the findings was carried out, exploring the information available in the studies.

This review shows a higher occurrence of hospitalization in males and with mean age > 60 years, in line with the literature, which has already shown that male sex and older age were risk factors for severe COVID-19 [6,44,45,46]. In addition, in line with the literature, we found a high prevalence of comorbidities in the sample, especially hypertension and diabetes [46,47]. Other comorbidities described in the literature had lower prevalence in the sample in this systematic review, for example, CKD (12.15%), cancer (7.26%), asthma (7.20%), COPD (6.83%) and immunosuppression (5.98%).

The prognosis of patients hospitalized with COVID-19 differed according to country and hospital setting. We observed high mortality in people hospitalized in hospital wards and ICU, which differed between countries. Morbidity, assessed as ARDS, had a high prevalence in hospital wards and ICU inpatients, evidencing the severity of the disease and the risk of a more serious prognosis. Moreover, it is already described in the literature that patients with ARDS present a higher proportion of comorbidities, including hypertension and diabetes [48], very prevalent in this review. Morbidity, assessed as MOFS, was reported for fewer studies and it seems to be a less prevalent condition. Regarding the prognosis of recovery, this systematic review demonstrated a high prevalence of fatal cases, with most patients hospitalized in hospital wards discharged to their homes (range 57–94%). In ICU, this prognosis was more heterogeneous, and this may be due to the high proportion of people that remain in ICU or are transferred to hospital wards.

In the methodological quality assessment, the risk of bias related to the control and presentation of possible confounders of the studies is highlighted. The heterogeneity of pharmacological and clinical therapies, as well as the presence of comorbidities, can lead to outcomes and should be reported impartially and transparently by researchers.

The design and methodological rigor can be considered a limitation of the evidence included in this review. Studies also did not adequately present follow-ups of participants to describe prognosis over time. Furthermore, the confounders of the studies were not reported accurately. The absence of pooled prognosis is highlighted as a limitation of the review, however, this is considered justified due to the heterogeneity of the included studies.

A previous systematic review provided evidence for the prognosis of COVID-19 in specific populations, for example, patients with obesity [49,50,51,52], acute kidney disease (AKI) [53,54], liver disease [54], vein thrombosis [55], cancer [56] and comorbidities in general [57]. All reviews found high mortality in COVID-19 patients. However, all published reviews were conducted with the inclusion of inadequate study designs for describing the prognosis, including retrospective and case–control studies, and no inception cohort was defined. In contrast to the previous reviews, our current evidence is the first systematic review with rigorous inclusion criteria and methodological rigor.

Our findings are important to provide information about the prognosis of COVID-19 and describe details of the conditions that can influence in the prognosis, important for public and clinical decision-making, through evidence of high methodological rigor. This study demonstrates the existing gap in the literature of methodologically adequate observational studies capable of describing the prognosis of COVID-19 more precisely and encouraging their execution.

We recommend that prospective inception cohort studies assessing prognosis in hospitalized patients with COVID-19 be conducted, with the definition of follow-up in time points (short, medium and long term), and control or transparent description and analysis of confounding factors, in addition to individual data being available for possible analyses, for example, survival analysis. Additionally, we recommend the use of a checklist for cohort studies of the Strengthening the Reporting of Observational Studies in Epidemiology (STROBE) statement [58].

## 5. Conclusions

The prognosis of people hospitalized with COVID-19 is characterized by negative morbidity and mortality outcomes, despite the fact that a proportion of the sample recovered. Furthermore, the prognosis varied depending on the hospital setting (severity), country and presence of comorbidities, emphasizing the need for appropriate prevention and management strategies. Future studies should be properly designed with adequate design aimed at exploring the COVID-19 prognosis over different time points, and to explore factors associated with outcomes.

## Figures and Tables

**Figure 1 ijerph-19-14609-f001:**
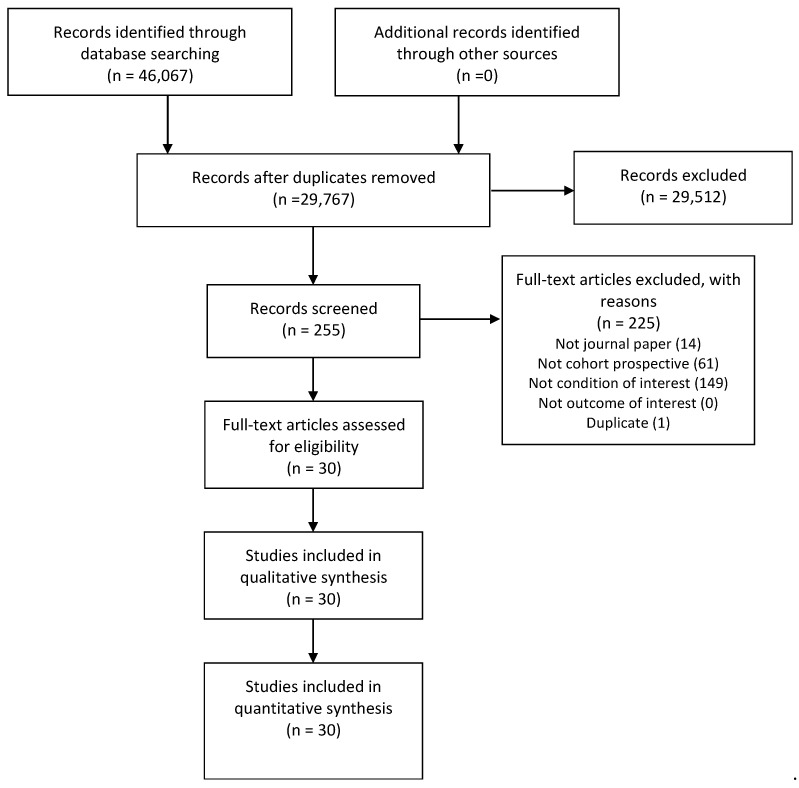
Flow of studies through the review. Potential full texts could be excluded for more than one reason.

**Figure 2 ijerph-19-14609-f002:**
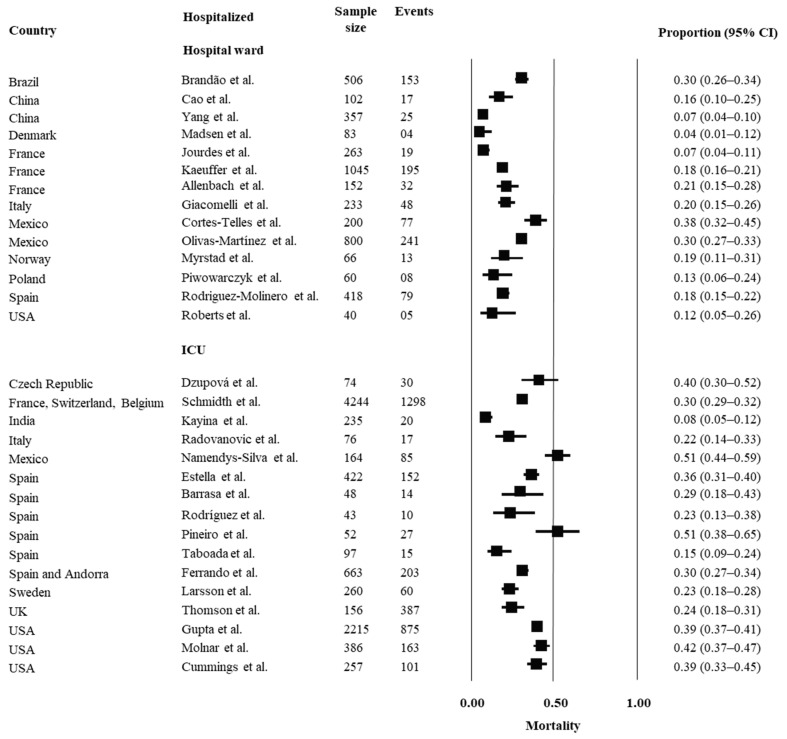
Prognosis of mortality in hospitalized people with COVID-19. For analysis, the studies were grouped by country and by hospital setting (hospital ward or ICU). Estimates of proportions for dichotomous data, considering the number of events and sample size, were reported. Abbreviations: USA = United States of America; UK = United Kingdom; IC = confidence interval. References: Brandão et al. [14]; Cao et al. [15]; Yang et al. [16]; Madsen et al. [17]; Jourdes et al. [18]; Kaeuffer et al. [19]; Allenbach et al. [20]; Giacomelli et al. [21]; Cortes-Telles et al. [22]; Olivas-Martínez et al. [23]; Myrstad et al. [24]; Piwowarczyk et al. [25]; Rodriguez-Molinero et al. [26]; Roberts et al. [27]; Dzupová et al. [28]; Schmidth et al. [29]; Kayina et al. [30]; Radovanovic et al. [31]; Namedys-Silva et al. [32]; Estella et al. [33]; Barrasa et al. [34]; Rodríguez et al. [35]; Pineiro et al. [36]; Taboada et al. [37]; Ferrando et al. [38]; Larsson et al. [39]; Thomson et al. [40]; Gupta et al. [41]; Molnar et al. [42]; Cummings et al. [43].

**Figure 3 ijerph-19-14609-f003:**
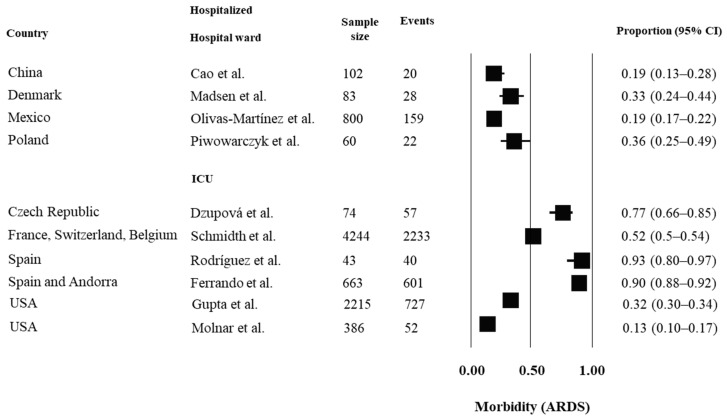
Prognosis of morbidity, considering ARDS, in hospitalized people with COVID-19. For analysis, the studies were grouped by country and by hospital setting (hospital ward or ICU). Estimates of proportions for dichotomous data, considering the number of events and sample size, were reported. Abbreviations: ARDS = acute respiratory distress syndrome; USA = United States of America; IC = confidence interval. References: Cao et al. [15]; Madsen et al. [17]; Olivas-Martínez et al. [23]; Piwowarczyk et al. [25]; Dzupová et al. [28]; Schmidth et al. [29]; Rodríguez et al. [35]; Ferrando et al. [38]; Gupta et al. [41]; Molnar et al. [42].

**Figure 4 ijerph-19-14609-f004:**
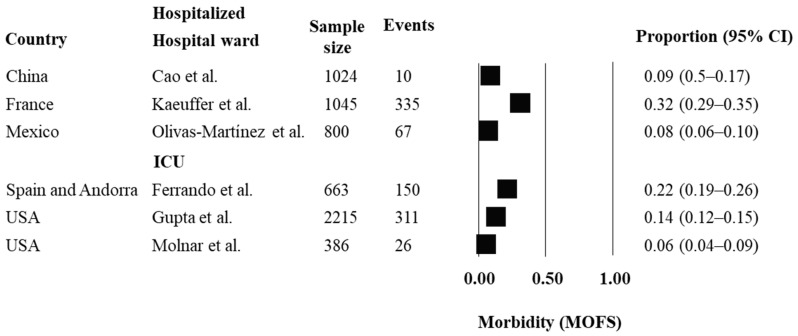
Prognosis of morbidity, considering MOFS, in hospitalized people with COVID-19. For analysis, the studies were grouped by country and by hospital setting (hospital ward or ICU). Estimates of proportions for dichotomous data, considering the number of events and sample size, were reported. Abbreviations: MOFS = multiorgan failure syndrome; USA = United States of America; IC = confidence interval. References: Cao et al. [15]; Kaeuffer et al. [19]; Olivas-Martínez et al. [23]; Ferrando et al. [39]; Gupta et al. [41]; Molnar et al. [42].

**Figure 5 ijerph-19-14609-f005:**
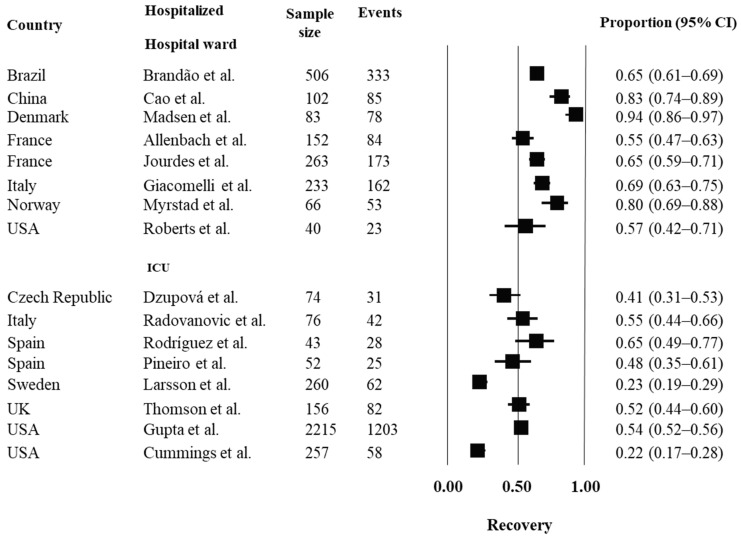
Prognosis of recovery in hospitalized people with COVID-19. For analysis, the studies were grouped by country and by hospital setting (hospital ward or ICU). Estimates of proportions for dichotomous data, considering the number of events and sample size, were reported. Abbreviations: ARDS = acute respiratory distress syndrome; USA = United States of America; UK = United Kingdom; IC = confidence interval. References: Brandão et al. [14]; Cao et al. [15]; Madsen et al. [17]; Allenbach et al. [20]; Jourdes et al. [18]; Giacomelli et al. [21]; Myrstad et al. [24]; Roberts et al. [27]; Dzupová et al. [28]; Radovanovic et al. [31]; Rodríguez et al. [35]; Pineiro et al. [36]; Larsson et al. [39]; Gupta et al. [41]; Thomson et al. [40]; Cummings et al. [43].

## Data Availability

All data generated or analyzed during this study are presented in the manuscript and Appendix A. Contact the corresponding author for access to the data presented in this study.

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
