# Peer review of "Recent Advance Analysis of Recovery in Hospitalized People with COVID-19: A Systematic Review"

_ijerph, 2022, doi:10.3390/ijerph192114609_

Round 1

Reviewer 1 Report (Previous Reviewer 1)

Dear respected Authors,

I have reviewed this search once before. The previous review had recommended some revision. On the recent version, researchers have done the review adequately.

I wish you all the best

Author Response

Responses to Reviewers

Manuscript ID: ijerph-2009676

REVIEWER COMMENTS:

REVIEWER #1

English language and style

( ) Extensive editing of English language and style required
( ) Moderate English changes required
(x) English language and style are fine/minor spell check required
( ) I don't feel qualified to judge about the English language and style

Dear respected Authors,

I have reviewed this search once before. The previous review had recommended some revision. On the recent version, researchers have done the review adequately.

I wish you all the best

Answer:

Thanks for the comment.

We inform that the study was revised and the adjustments are inserted throughout the text.

Reviewer 2 Report (Previous Reviewer 2)

In Figure 4, 10/1024 = 0.00977, not 0.09. The authors should carefully examine all numbers in all figures.  

Author Response

Responses to Reviewers

Manuscript ID: ijerph-2009676

REVIEWER COMMENTS:

REVIEWER #2

English language and style

( ) Extensive editing of English language and style required
( ) Moderate English changes required
(x) English language and style are fine/minor spell check required
( ) I don't feel qualified to judge about the English language and style

Comments and Suggestions for Authors

In Figure 4, 10/1024 = 0.00977, not 0.09. The authors should carefully examine all numbers in all figures.  

Answer:

Thanks for the comment. We made all the requested adjustments. Please, see Figure 4. In Figure 4, 10/102 = 0.0980, not 1024.

All numbers in all figures were carefully examined.

We inform that the study was revised and the adjustments are inserted throughout the text.

This manuscript is a resubmission of an earlier submission. The following is a list of the peer review reports and author responses from that submission.

Round 1

Reviewer 1 Report

TITLE

-    The title ‘Recent advance analysis of recovery in hospitalized people with COVID-19: a systematic review’ is concise and the main idea is clear.

ABSTRACT

-    The usage of a comprehensive Meta-Analysis to identify factors present through COVID-19 hospitalization should be revealed in the abstract.

-    Confidence interval (CI) is preferable to be added to percent sign %

-    Conclusions need to be substantiated or justified by the statistical analysis.

-    In the ABSTRACT, in the second sentence, please change the word (occured) to (occurred).

INTRODUCTION

-    In a general way, previous pertinent literature was cited and discussed.

-    Research hypotheses were not stated.

METHODS

-    For risk of bias (methodological quality) did you investigate the inter-rater reliability? Please illuminate that.

-    Doubts should be resolved beyond discussion.

RESULTS

-     The Figures are suitable.

-     General issues easily illustrated scientific information.

-    A pooled estimate was not calculated due to the heterogeneity of studies.

-    The pooled estimate had to be obtained, regardless heterogeneity.

DISCUSSION

-    Please start the Discussion with a short sentence explaining the most important findings of the study like “The most important finding of the present study was…”. 

CONCLUSIONS

-    The conclusions of the study should be entirely authenticated or justified by the statistical analysis.

Thank you

Author Response

Answers to Reviewers

Manuscript number: ijerph-1790186

REVIEWER COMMENTS:

REVIEWER #1

TITLE

-    The title ‘Recent advance analysis of recovery in hospitalized people with COVID-19: a systematic review’ is concise and the main idea is clear.

Answer:

Thank you for the approval.

ABSTRACT

Q1.    The usage of a comprehensive Meta-Analysis to identify factors present through COVID-19 hospitalization should be revealed in the abstract.

A1. Thanks for this specific comment. A sentence has been added, as requested (page 2, line 11-12).

We have included the use of a comprehensive Meta-Analysis in the abstract to identify variables present during COVID-19 hospitalization, as follows:

Methods: MEDLINE, EMBASE, AMED, and COCHRANE databases were searched for studies published up to June 28th, 2021 without language restrictions. Descriptors were related to “COVID-19” and “prognosis”. Prospective inception cohort studies that assessed morbidity, mortality and recovery in hospitalized people over 18 years old with COVID-19 were included. Two independent reviewers selected eligible studies and extracted available data. Acute respiratory distress syndrome (ARDS) and multiple organ failure (MOFS) were considered as outcomes for morbidity and discharge was considered for recovery. The descriptive analyses and data summarization were performed using Comprehensive Meta-Analysis (version 2.2.064). The Quality in Prognosis Studies (QUIPS) tool was used to assess risk of bias. Raw data were analysed descriptively.

Q2.    Confidence interval (CI) is preferable to be added to percent sign %

A2. We have choosen to provide the percentage range of the key findings in the abstract in order to keep it concise. We point out the fact that the confidence interval (CI) was fully specified in the body of the manuscript (please see figures 2-5).

Q3.    Conclusions need to be substantiated or justified by the statistical analysis.

A3. Using Comprehensive Meta-Analysis, the results were presented in this study in a descriptive manner. Notably, the results cannot extrapolate the analysis conducted because the target of the present study was to describe prognosis in hospitalized adults with COVID-19. Accordingly, it was possible to draw the following conclusion from our analysis: the prognosis of patients hospitalized with COVID-19 varied in relation to the hospital setting (severity), country, and prevalence of comorbidity, indicating the need for increased attention to prevention strategies and for both public and clinical decision-making.

Q4.    In the ABSTRACT, in the second sentence, please change the word (occured) to (occurred).

A4. Done as requested. Please see (page 2, line 2 in the abstract).

Q5. INTRODUCTION

-    In a general way, previous pertinent literature was cited and discussed.

-    Research hypotheses were not stated.

A5. Thanks for the comment. Please see the research hypotheses below:

The hypothesis of the present study was that the outcomes occurring in COVID-19 hospitalized patients could be impacted by factors such as comorbidity prevalence, hospital setting (severity), and country.

Q6. METHODS

-    For risk of bias (methodological quality) did you investigate the inter-rater reliability? Please illuminate that.

A7.  Hayden et al. (2013) and Grooten et al. (2014) assessed the QUIPS’ inter-rater reliability (2019). Additionally, a pilot research was conducted, and the procedures were followed in accordance with Cochrane Recommendations, as described in the following sentence:

In the present systematic review, two independent reviewers (JNS and ACF), with the same degree of training to use the form beforehand, extracted data from the included studies and a third reviewer (LBM) resolved disagreements. In addition, two independent reviewers (JNS and ACF) assessed the methodological quality of the included studies using Quality in Prognosis Studies (QUIPS) tool modified [10,11,12]. A third reviewer (LBM) resolved disagreements. Reviewers used a standardized form downloaded from the Cochrane Methods Prognosis website [13] and received the same level of training to use the form a priori.

Q7.    Doubts should be resolved beyond discussion.

A7. We answered this query with our previous answers.

Q8. RESULTS

-     The Figures are suitable.

-     General issues easily illustrated scientific information.

-    A pooled estimate was not calculated due to the heterogeneity of studies.

-    The pooled estimate had to be obtained, regardless heterogeneity.

A8. Thanks for the appreciation of our Figures and scientific information.  This systematic review was reported according to PRISMA checklist and Cochrane Recommendations [7,8]. Finally, we emphasize that Cochrane does not recommend meta-analysis when the studies present great heterogeneity.

Q9. DISCUSSION

-    Please start the Discussion with a short sentence explaining the most important findings of the study like “The most important finding of the present study was…”. 

A9. Thanks for the comment. We have included in the beginning of the study the following sentence:

The most important finding of the present study was that the people hospitalized with COVID-19 manifested negative outcomes, morbidity and mortality, despite a proportion of the sample resulting recovered. Of note, this was the first systematic review based on a rigorous methodological design with inclusion of prospective cohort studies and with defined inception cohort, following Cochrane Recommendations [8,13], carried out in order to describe the prognosis in morbidity, mortality, and recovery outcomes in people hospitalized with COVID-19.

Q10. CONCLUSIONS:    The conclusions of the study should be entirely authenticated or justified by the statistical analysis.

A10.  Since the data were presented descriptively using Comprehensive Meta-Analysis, the conclusion cannot extrapolate the analysis performed. Thus, it was possible to conclude:

Prognosis of people hospitalized with COVID-19 present occurrence of negative outcomes, morbidity and mortality, despite a proportion of the sample resulting recovered. Furthermore, prognosis varied in relation to the hospital setting (severity), country and prevalence of comorbidity, pointing to the necessity of increasing attention for prevention strategies and for public and clinical decision-making. Future studies should be properly designed with adequate design to explore COVID-19 prognosis, over different time points, and to explore factors associated with outcomes.

Reviewer 2 Report

In this paper, the authors performed a meta-analysis of the recovery of COVID-19 patients. Data from previously published 30 cohort studies were used in this study. The main founding of this paper are the prognosis of morbidity with/without other comorbidities and recovery proportion among hospitalized patients. All previously published studies used in this meta-analysis were conducted in the first half of 2020. Although a great deal of work was performed, lacking details in the method section need to be carefully revised before publication.

1.       In the data analysis section, the authors only mentioned that the Comprehensive Meta-Analysis was used for the data analysis. I assume the “Comprehensive Meta-Analysis” is a software, right? If so, which version of this software was used in this study? If not, more details of the comprehensive meta-analysis need to be provided to help the readers to understand the corresponding procedure. Also, add corresponding citations for this.

2.       Figure is nice for the readers to quickly capture the data filter procedure, it will be better to add corresponding data analysis in this figure, corresponding to the method sections.

3.       In these figures, almost no negative values, the x-axis from -1 to 0 should be removed in the figure. Also, the authors should be carefully examining the data labeled in the figure, for example, in Figure 4, the data of China is not correctly labeled. 10/1024 = 0.00976, not 0.09.

4.       In table S1, the sample size from Schmidth et al. study is 4.244. This needs to be corrected.

Author Response

Answers to Reviewers

Manuscript number: ijerph-1790186

REVIEWER COMMENTS:

REVIEWER #2

Q1. In this paper, the authors performed a meta-analysis of the recovery of COVID-19 patients. Data from previously published 30 cohort studies were used in this study. The main finding of this paper is the prognosis of morbidity with/without other comorbidities and recovery proportion among hospitalized patients. All previously published studies used in this meta-analysis were conducted in the first half of 2020. Although a great deal of work was performed, lacking details in the method section need to be carefully revised before publication.

A1. Thanks for the appreciation of our work. We have revised and improved all the methods section, as recommended.

Q2.       In the data analysis section, the authors only mentioned that the Comprehensive Meta-Analysis was used for the data analysis. I assume the “Comprehensive Meta-Analysis” is a software, right? If so, which version of this software was used in this study? If not, more details of the comprehensive meta-analysis need to be provided to help the readers to understand the corresponding procedure. Also, add corresponding citations for this.

A2. We have inserted additional information in the manuscript in order to clarify the above-cited (see page 5, line 15). Please see below:

The descriptive analyses and data summarization were performed using Comprehensive Meta-Analysis (version 2.2.064).

Q3. Figure is nice for the readers to quickly capture the data filter procedure, it will be better to add corresponding data analysis in this figure, corresponding to the method sections.

A3. We have added the corresponding data analysis in the figures as follows:

Figure 2. Prognosis of mortality in hospitalized people with COVID-19. For analysis, the studies were grouped by country and by hospital setting (hospital ward or ICU). Estimates of proportions for dichotomous data, considering number of events and sample size, were reported. Abbreviations: USA=United States of America; UK=United Kingdom; IC=confidence interval.

Figure 3. Prognosis of morbidity, considering ARDS, in hospitalized people with COVID-19. For analysis, the studies were grouped by country and by hospital setting (hospital ward or ICU). Estimates of proportions for dichotomous data, considering number of events and sample size, were reported. Abbreviations: ARDS=acute respiratory distress syndrome; USA=United States of America; IC=confidence interval.

Figure 4. Prognosis of morbidity, considering MOFS, in hospitalized people with COVID-19. For analysis, the studies were grouped by country and by hospital setting (hospital ward or ICU). Estimates of proportions for dichotomous data, considering number of events and sample size, were reported. Abbreviations: MOFS=multiorgan failure syndrome; USA=United States of America; IC=confidence interval.

Figure 5. Prognosis of recovery in hospitalized people with COVID-19. For analysis, the studies were grouped by country and by hospital setting (hospital ward or ICU). Estimates of proportions for dichotomous data, considering number of events and sample size, were reported. Abbreviations: ARDS=acute respiratory distress syndrome; USA=United States of America; UK=United Kingdom; IC=confidence interval.

Figure S1. Prevalence of hypertension in the sample. For analysis, the studies were grouped by country and by hospital setting (hospital ward or ICU). Estimates of proportions for dichotomous data, considering number of events and sample size, were reported. Abbreviations: ICU=intensive care unit; CI=confidence interval; USA=United States of America; UK=United Kingdom.

Figure S2. Prevalence of diabetes in the sample. For analysis, the studies were grouped by country and by hospital setting (hospital ward or ICU). Estimates of proportions for dichotomous data, considering number of events and sample size, were reported.  Abbreviations: ICU=intensive care unit; CI=confidence interval; USA=United States of America; UK=United Kingdom.

Figure S3. Prevalence of obesity in the sample. For analysis, the studies were grouped by country and by hospital setting (hospital ward or ICU). Estimates of proportions for dichotomous data, considering number of events and sample size, were reported. Abbreviations: ICU=intensive care unit; CI=confidence interval.

Figure S4. Prevalence of smoking history in the sample.  For analysis, the studies were grouped by country and by hospital setting (hospital ward or ICU). Estimates of proportions for dichotomous data, considering number of events and sample size, were reported. Abbreviations: ICU=intensive care unit; CI=confidence interval; USA=United States of America.

Q4. In these figures, almost no negative values, the x-axis from -1 to 0 should be removed in the figure. Also, the authors should be carefully examining the data labeled in the figure, for example, in Figure 4, the data of China is not correctly labeled. 10/1024 = 0.00976, not 0.09.

A4. Thanks for the comment. We have made all the requested adjustments. Please, see Figures 2-5 and Figure S1-S4.

Q5. In table S1, the sample size from Schmidth et al. study is 4.244. This needs to be corrected.

A5. Thanks for the comment. We have made the requested adjustment. As requested, we have modified the point for comma.
